REGISTERED REPORT PROTOCOL

# Exploring the impact of trait number and type on functional diversity metrics in real-world ecosystems

Timothy Ohlert[1]*, Kaitlin Kimmel[2,3], Meghan Avolio[3], Cynthia Chang[4], Elisabeth Forrestel[5], Benjamin Gerstner[1], Sarah E. Hobbie[6], Kimberly Komastu[7], Peter Reich[8,9,10], Kenneth Whitney[1]

1 Department of Biology, University of New Mexico, Albuquerque, NM, United States of America, 2 Mad Agriculture, Boulder, Colorado, 3 Department of Earth & Planetary Sciences, Johns Hopkins University, Baltimore, MD, United States of America, 4 Division of Biological Sciences, University of Washington, Bothell, WA, United States of America, 5 Department of Viticulture and Enology, University of California, Davis, CA, United States of America, 6 Ecology, Evolution and Behavior Department, University of Minnesota, St. Paul, MN, United States of America, 7 Smithsonian Environmental Research Center, Edgewater, MD, United States of America, 8 Department of Forest Resources, University of Minnesota, Minneapolis, MN, United States of America, 9 Institute for Global Change Biology and School for Environment and Sustainability, University of Michigan, Ann Arbor, MI, United States of America, 10 Hawkesbury Institute for the Environment, Western Sydney University, Penrith South, NSW, Australia

* tohlert@unm.edu

This is a Registered Report and may have an associated publication; please check the article page on the journal site for any related articles.

## Abstract

The use of trait-based approaches to understand ecological communities has increased in the past two decades because of their promise to preserve more information about community structure than taxonomic methods and their potential to connect community responses to subsequent effects of ecosystem functioning. Though trait-based approaches are a powerful tool for describing ecological communities, many important properties of commonly-used trait metrics remain unexamined. Previous work in studies that simulate communities and trait distributions show consistent sensitivity of functional richness and evenness measures to the number of traits used to calculate them, but these relationships have yet to be studied in actual plant communities with a realistic distribution of trait values, ecologically meaningful covariation of traits, and a realistic number of traits available for analysis. Therefore, we propose to test how the number of traits used and the correlation between traits used in the calculation of functional diversity indices impacts the magnitude of eight functional diversity metrics in real plant communities. We will use trait data from three grassland plant communities in the US to assess the generality of our findings across ecosystems and experiments. We will determine how eight functional diversity metrics (functional richness, functional evenness, functional divergence, functional dispersion, kernel density estimation (KDE) richness, KDE evenness, KDE dispersion, Rao's Q) differ based on the number of traits used in the metric calculation and on the correlation of traits when holding the number of traits constant. Without a firm understanding of how a scientist's choices impact these metric, it will be difficult to compare results among studies with different metric parametrization and thus, limit robust conclusions about functional composition of communities across systems.

**Data Availability Statement:** All relevant data from this study will be made available upon study completion.

**Funding:** KW - DEB-1257965 National Science Foundation Division of Environmental Biology https://www.nsf.gov/div/index.jsp?div=DBI SH - DBI- 1725683, DEB-1753859, DEB- 1831944 National Science Foundation Division of Environmental Biology, National Science foundation Division of Biological Infrastructure https://www.nsf.gov/div/index.jsp?div=DBI PR - DBI- 1725683, DEB-1753859 National Science Foundation Division of Environmental Biology, National Science foundation Division of Biological Infrastructure https://www.nsf.gov/div/index.jsp?div=DBI EF - DEB-0841917 National Science Foundation Division of environmental biology https://www.nsf.gov/div/index.jsp?div=DBI The funders had no role in study design, data collection and analysis, decision to publish, or preparation of the manuscript.

**Competing interests:** The authors have declared that no competing interests exist

## Introduction

Trait-based diversity measures have advanced the field of community ecology by increasing our understanding of both community assembly and diversity impacts on ecosystem functions [1, 2]. Functional diversity metrics allow researchers to quantify multiple facets of diversity, place an emphasis on mechanisms of community assembly, and provide a 'common currency' by which communities can be compared across sites and ecosystems [3, 4].Traditional measures for characterizing communities, such as species richness and species ordinations, use species' taxonomic classifications as discrete units, but functional diversity metrics can preserve more information about community assembly and function by including traits of species organized on continuous axes [5, 6].

Several aspects of functional and taxonomic diversity have been extensively studies. Scientists have probed functional diversity's correlation with species richness [7, 8] and ecosystem functioning [4], the importance of intraspecific trait variation for diversity [4, 9, 10], and the ecological hypotheses that functional diversity metrics can test, such as optimal strategies or functional turnover [6, 11]. Many taxonomic measures of community diversity have been extensively studied for their mathematical properties to allow these metrics to be comparable across sites and ecosystems, such as Shannon's diversity and Simpson's evenness that have mathematical characteristics linked to species number [12, 13]. Similarly, functional diversity metrics have mathematical characteristics that may cause the number or type of traits used to calculate the metric to impact the measure. For example, multidimensional metrics are calculated with additional dimensions for each additional trait included, and the correlation between traits affects the importance of each dimension to the metric. Therefore, functional diversity could differ among replicate plots or sites simply because of the number or types of traits used to calculate the metric without any underlying ecological basis. Though single-trait indices are an effective tool for linking trait diversity to specific ecosystem processes [14, 15], indices based on multiple traits may better match ecological theories of community assembly around multidimensional niche space [16–18]. As use of multi-trait functional diversity increases, it is important to determine the conditions under which they reflect ecological processes as opposed to mathematical patterns.

Studies using simulated communities have tested whether the number and correlation of traits used in functional diversity metrics can impact the magnitude of the metric [7, 19]. Using simulated data, Legras et al. [19] showed that functional richness and functional divergence metrics decreased with increased trait number, but functional evenness metrics were not responsive to increasing trait numbers. Also using simulated data, Cornwell et al. [7] showed that convex hull volume (commonly referred to as "functional richness") tended to decrease with increasing correlation among traits included in the metric calculation, and that the decrease was greater in more species-rich communities. The limitations of functional diversity metrics described in these studies with simulated community data could be exacerbated when applied in natural communities. Calculating functional diversity measures in natural communities poses additional challenges both ecological and practical. Real plant communities are non-random assemblages of species which are influenced by competitive interactions, coexistence, mutualisms, niche partitioning, and environmental filtering among many other processes of community assembly [20–25]. Functional diversity metrics are likely to exhibit patterns due to ecologically meaningful correlation of traits in real communities, in particular, among suites of traits typically used in community ecology such as the leaf economic spectrum and root economic spectrum [26, 27]. Moreover, real data collection introduces constraints on trait data, such as realistic numbers of traits collected given limited resources and missing trait data, particularly for rare species. Functional diversity metrics,

therefore, are most often calculated with fewer traits and fewer species than those in studies based on simulated communities.

The field lacks clear guidelines for researchers to follow when choosing the number and types of traits to include when calculating functional diversity metrics. Decisions are often based on researcher intuition and the practices of similar studies, but such intuition and interpretation of trait selection can be improved by rigorous exploration of the impact of trait selection on diversity metrics [4, 28, 29]. These decisions can fall along a spectrum of options ranging from selecting the minimum number of traits needed to calculate a metric to using every trait available. For example, some studies suggest that researchers use a small number of traits related to certain ecosystem properties or other topics of interest (e.g., [8]), regardless of how correlated they may be. Other studies use all available traits in order to maximize the dimensions of diversity being studied in an effort to comprehensively assess the niche space that species and communities occupy (e.g., [30]). Choosing traits that are highly correlated can result in an underrepresentation of the diversity of functions present by overemphasizing groups of traits which describe similar processes, such as traits involved in the leaf economics spectrum [31]. Further, functional diversity metric calculation in high dimensional space can require dimensionality reduction–another decision that can impact the value of the metric calculated. However, few studies scrutinize how these decisions can impact conclusions when using functional diversity metrics to characterize communities.

Here, we aim to understand how the number of traits and correlation between traits impact functional diversity values. We will focus on eight measures of functional diversity that express principal facets of community trait composition (see *Table 1* for more details on each metric): functional richness (FRich), functional evenness (FEve), functional divergence (FDiv), functional dispersion (FDis), Rao's Q, kernel density estimation (KDE) richness, KDE evenness, and KDE dispersion [31–34]. We will use trait data from real (natural/intact and experimental) plant communities, which will allow us to understand how these metrics respond to a realistic spread of traits and species richness. In this study, we will use trait data collected from three U. S. grasslands, which range from tallgrass prairie to desert grassland, to test impacts of trait number and identity in functional diversity metric values. Our dataset includes plant traits collected on location at these three sites that include both naturally assembled and planted communities.

Specifically, we ask:

1. Do functional richness, functional evenness, and functional dispersion vary with respect to the number and correlation of traits used? Based on findings from [19], we expect functional richness, KDE richness, functional dispersion, and functional divergence to decrease with increasing numbers of traits, but for Rao's Q to increase [35] and functional evenness to be unresponsive to the number of traits. We do not have *a priori* hypotheses for KDE evenness and KDE dispersion since properties of these metrics have yet to be explicitly studied. Based on [7], we expect that functional richness will be greater when traits are less correlated. However, we do not have directional hypotheses for the rest of the metrics.

2. Is metric sensitivity to trait number/type consistent across sites and experiments? If metric sensitivity is consistent across sites, it will be easier to standardize functional diversity metrics across different studies. If sensitivity is not consistent across sites, further investigation will be necessary to understand the consequences of this when comparing functional diversity across sites.

**Table 1.**

| Functional diversity metric | Abbreviation | Ecological relevance | Examples of usage | Citations |
|---|---|---|---|---|
| Functional richness | FRich | Functional space filled by the community | De Vries and Bardgett 2016 [50]<br>De la Riva et al. 2018 [51]<br>Lourenco Jr. et al. 2021 [52] | Cornwell et al. 2006 [7], Villeger et al. 2008 [8] |
| Kernel density richness | KDE richness | Functional space filled by the community | Soares et al. 2022 [53]<br>Piano et al. 2020 [54]<br>Pavlek & Mammola 2020 [55] | Blonder 2018 [47], Mammola and Cardosso 2020 [34] |
| Functional evenness | FEve | The similarity trait abundances within the community | De bello et al. 2012 [56]<br>Niu et al. 2016 [57]<br>Biswas et al. 2019 [58] | Villeger et al. 2008 [8] |
| Kernel density evenness | KDE evenness | Similarity of trait abundances within the community | Soares et al. 2022 [53]<br>Piano et al. 2020 [54] | Mammola and Cardosso 2020 [34] |
| Functional dispersion | FDis | Average trait difference between individuals within the community | Zuo et al. 2021 [59]<br>Shovon et al. 2019 [60]<br>Griffin-Nolan et al. 2019 [61] | Laliberte and Legendre 2010 [33] |
| Functional divergence | FDiv | Average trait difference between individuals within the community | Janschke et al. 2019 [62]<br>Ebeling et al. 2017 [63]<br>Thakur & Chawla 2019 [64] | Villeger et al. 2008 [8] |
| Rao's quadratic entropy | Rao's Q | Average trait difference between individuals within the community | De Bello et al. 2009 [65]<br>Ebeling et al. 2014 [66]<br>Pillar et al. 2013 [67]<br>Wang et al. 2018 [68] | Rao 1982 [70], Botta-Dukat 2005 [46] |
| Kernel density dispersion | KDE dispersion | Average trait difference between individuals within the community | Piano et al. 2020 [54]<br>Greenop et al. 2021 [69] | Mammola and Cardoso 2020 [34] |

## Methods

### Site descriptions

Here we will use data from three grassland sites across the United States that span a range of climate (MAP 250mm—866 mm, MAT 6˚C—15˚C) and species diversity. We will use two sites with naturally assembled communities and one with a planted community in order to be representative of the state of grassland studies where some use naturally assembled communities while others use planted communities.

Cedar Creek Ecosystem Science Reserve (East Bethel, Minnesota, USA) is in central Minnesota and classified as a tallgrass prairie. According to Koppen and Geiger classification, the climate is characterized as cold continental with hot summer, but without a dry season [Peel 2007]. The mean growing season (May–August) precipitation is approximately 420 mm, mean minimum growing season temperature is 12˚C, and mean maximum growing season temperature is 25˚C (1982–2016 period; http://www.cedarcreek.umn.edu/research/data). Soils at Cedar Creek are characterized as nutrient-poor entisols derived from a glacial outwash sand plain [36]. The study from Cedar Creek consists of artificially planted communities.

Konza Prairie Biological Station (Manhattan, Kansas, USA) is in eastern Kansas in the Flint Hills ecoregion. Konza is classified as a tallgrass prairie, and much of the site has remained unplowed throughout its history [37]. Konza's growing season extends from roughly May-October, with annual precipitation averaging 835 mm and average July air temperature of 27C [37].

The Sevilleta National Wildlife Refuge is in central New Mexico at the northern edge of the Chihuahuan Desert. The Sevilleta includes desert grasslands, and the climate is characterized as cold semi-arid according to the Koppen and Geiger classification [36]. The growing season is characterized by two rainy periods (March—May and July—September) split by a dry

period. The mean monsoon growing season precipitation is approximately 150 mm and the mean monsoon growing season temperature is 22C.

## Community composition data

We will use one to four studies at each site (n = 7 studies total) within one year to characterize the functional diversity of grassland plant communities.

At Cedar Creek, we will use community composition data from all 16-species plots in a bio-diversity, $CO_2$, and nitrogen addition experiment (BioCON, n = 48). All 16-species plots were originally planted with the same mixture of species (*Achillea millefolium*, *Amorpha canescens*, *Andropogon gerardii*, *Anemone cylindrica*, *Asclepias tuberosa*, *Bouteloua gracilis*, *Bromus inermis*, *Elymus repens*, *Koeleria cristata*, *Lespedeza capitata*, *Lupinus perennis*, *Petalostemum villosum*, *Poa pratensis*, *Schizachyrium scoparium*, *Solidago rigida*, and *Sorghastrum nutans*) such that all species were seeded at the same density in 1997. Plots are weeded every year to remove invading species. Through time, the plots can lose species (and regain those), but could never gain new species. Further, species abundances shifted from the equal proportion planted in the first year. Every August, species abundances were visually estimated in a 1 m$^2$ permanent plot. Here, we used data from 2020—the most recent year species abundances are available.

At both Konza and Sevilleta, we will use several studies as representative plant communities. This ensures that we will at least have one study per site if we need to drop observations because trait coverage is too low (see below for discussion).

At Konza, we will use community composition data from 4 watershed transects with different burn frequencies and grazing patterns. Konza is dominated by a few C4 grass species (*Andropogon gerardii*, *Schizachyrium scoparium*, *Sorghastrum nutans*), with the bulk of species diversity made up by C3 grasses and forbs [37]. Specifically, we will use one watershed that was burned annually but never grazed, one that was burned annually and grazed, one that was burned every 20 years but never grazed, and one that was burned every 20 years and grazed. Cover was estimated in permanent 1x1m plots twice per year. We will use the maximum cover of the species between these two sampling times to get a cover estimate per species. We will use data from the 2010 sampling because it was the same year that trait data were collected at Konza.

At Sevilleta, we used community composition data from two observational sites, one in a Great Plains grassland ecosystem and the other in a desert grassland ecosystem. The Great Plains grassland is dominated by *Bouteloua gracilis* (blue grama), a long-lived, caespitose, C4 perennial grass common throughout much of the United States and Canada. The desert grassland is dominated by *Bouteloua eriopoda* (black grama), a stoloniferous C4 perennial grass common in the southwestern United States and Mexico. These two dominant perennial grasses account for about 80% of vegetative cover in their respective ecosystems. Each site has 30 1x1m quadrats which were assayed in September of 2018, at the peak of the post-monsoon growing season and around the same time that trait data were collected. In each quadrat, plants were identified to species and their percent ground cover was visually estimated.

## Trait data

Trait data were collected for the individuals found at each of the different sites. Thus, our trait data are representative of the traits actually found in the given community and not just an average independent of location. Traits include measurements from leaves (e.g. specific leaf area), stems (e.g. stem dry matter content), roots (e.g. root dry matter content), whole-plant (e.g. height), and ecological attributes (e.g. amount of nitrogen in monoculture). Including traits across these measurement categories provides a more-complete representation of

community assemblages [38–41]. For detailed descriptions of trait collection protocols at each site, see S1 File.

At Cedar Creek, we will use trait data collected in the monoculture plots of the BioCON experiment. We will use trait data from monoculture plots that correspond to the $CO_2$ and N treatments to match with 16-species community plots. Data were collected between 1998 and 2020. Some traits were collected over multiple years whereas others were only collected once. In total, there were 10 distinct traits: specific leaf area (SLA), $I^*$ (the amount of light at the soil surface in monoculture), $R^*$ (the amount of nitrogen in monoculture), root %C, root %N, total root biomass, shoot %N, shoot %C, and seed mass.

At Konza, we will use trait data collected in a watershed that was burned annually and had no grazers. In total there were 12 distinct traits: plant height, leaf area, specific leaf area, leaf dry matter content, stomatal length, stomatal density, stomatal pore area index, leaf %N, leaf % C, d13C, photosynthetic pathway, and growth form.

At Sevilleta, we will use trait data collected primarily from September to November of 2017 on individuals growing under ambient conditions near permanent ambient plots used to monitor plant communities. The full suite of traits were often measured on the same individuals, up to 10 individuals per species. In total there were 10 distinct traits: maximum plant height, leaf dry matter content, specific leaf area, d15N, d13C, leaf %N, leaf %C, stem dry matter content, root dry matter content, and photosynthetic pathway.

For each trait at each site, we will calculate an average trait value based on all the measurements for the given species and trait. We acknowledge that this obscures variation within a given trait (intraspecific variation) for a species; such variation can be quite important for some questions [10, 42–44]. The impacts of intraspecific variation in this study are minimized by only using trait values collected at each site, but sufficient data were not collected for each trait of each species to include intraspecific variation into our analysis. Before analysis, we will remove species that have less than 100% trait coverage. We will, however, make sure that the communities are still represented by at least 80% of species abundance–this approach de-emphasizes the importance of rare species, but is a logistical constraint faced by many researchers doing trait analyses. This will ensure that we are representing the community to the best of our ability with the given trait data.

## Brief background on functional diversity metrics

We will focus our analyses on eight common functional diversity metrics: functional richness (FRich) [8], functional evenness (FEve) [9], functional dispersion (FDis), functional divergence (FDiv), Rao's Q, kernel density estimation (KDE) richness, KDE evenness, and KDE dispersion [33]. FRich is the multidimensional equivalent of a range [8]. It is calculated as the convex hull volume that is made from all trait values for up to $n$ traits in the community. The number of dimensions used to calculate the final volume can be reduced from the total trait number [45]. FEve is the minimum spanning tree to quantify the regularity of branch lengths and the evenness in trait relative abundances. For each branch, $l$, of the minimum spanning tree, the weighted evenness ($EW$) is calculated as $EW_l = \frac{dist(i,j)}{w_i + w_j}$ where $i$ and $j$ are species, and $w_i$ is the relative abundance of species $i$. Then, the partial weighted evenness ($PEW$) is then calculated for each branch as $PEW_l = \frac{EW_l}{\sum_{l=1}^{S-1} EW_l}$, where $S$ is the total number of species in the community. FEve is then defined as $\frac{\sum_{l=1}^{S-1} \min\left(PEW_l, \frac{1}{S-1}\right) - \frac{1}{S-1}}{1 - \frac{1}{S-1}}$ [9]. FDis is the weighted mean distance between species and a weighted-centroid. It is calculated as $\frac{\sum a_j z_j}{\sum a_j}$ where $a_j$ is the relative

abundance of species $j$ and $z_j$ is the distance species $j$ is from the weighted centroid [33]. FDiv is a relative abundance-weighted spread of traits along a trait axis independent of functional richness and is calculated as $\frac{\Delta d + \overline{dG}}{\Delta|d| + \overline{dG}}$ where $\overline{dG}$ is the mean distance of species to the weighted-centroid and $\Delta d$ is the sum of relative abundance-weighted deviances from the weighted-centroid [9]. Rao's Q measures the pairwise differences in traits between species in a community and is calculated as $\sum_{i-1}^{S-1} \sum_{j=i+1}^{S} d_{ij} p_i$ where $S$ is the number of species in the community, $d_{ij}$ is the functional difference between the $i$-th and $j$-th species, and $p$ is a vector of relative abundance values [46]. These five functional diversity metrics commonly incorporate distance measures by reducing dimensionality using principal coordinates analysis (PCoA) to return PCoA axes which are used to calculate the functional diversity metrics. However, we will avoid this dimensionality reduction when possible (for all metrics except FRich, see discussion in *Functional Diversity Calculations* section). *n*-dimensional hypervolumes use Gaussian kernel density estimation (KDE) to create a relative abundance-weighted probability distribution of traits in multidimensional space [34]. All KDE-based functional diversity metrics will be calculated using the hypervolume and bat packages in R [34, 47]. KDE richness is the total volume of the *n*-dimensional hypervolume created from unweighted trait values present in the community. KDE evenness is the overlap between the abundance-weighted *n*-dimensional hypervolume and a similar hypervolume in which all traits and abundances are distributed evenly. KDE dispersion is the average distance between random points within the *n*-dimensional hypervolume and the hypervolume centroid.

## Functional diversity calculations

For each site, we will follow the same protocol for calculating functional diversity metrics. We will calculate FRich, FEve, and FDis, FDiv, and Rao's Q using the FD package in R [45] using both Gower and Euclidean dissimilarity as the distance measure, along with using the hypervolume package in R to calculate KDE *n*-dimensional hypervolumes which are passed to the bat package to create KDE richness, KDE evenness, and KDE dispersion [34, 47]. Gower dissimilarity has the capacity to calculate distances with categorical traits, though Euclidean dissimilarity is better for continuous traits. Functional diversity metrics from the FD package and kernel density estimation are among the most-used metrics for quantifying trait-based diversity within communities due to both ease of use and ecological relevance [34, 45]. We will use dimensionality reduction where necessary in our analyses. First, PCoA is done on the species-species matrix for each set of traits we consider. The categorical variables are taken into consideration in the creation of the distance matrix which is done using Gower dissimilarity. Thus, the PCoA is done on the continuous distance values rather than on the raw traits. Similar to Legras et al. [19], we are going to hold the number of dimensions equal to 2 for only our calculation of functional richness (FRich) as the other metrics do not require dimensionality reduction (note: FRich does not either if all traits are continuous, but we have several categorical trats in our dataset). We will then conduct a sensitivity analysis to determine if holding the number of dimensions equal to 3, 4, and the maximum (dimensions = number of traits when using all continuous traits or dimensions = number of traits-1 when including categorical traits) produce similar results. Each metric uses species presence/absence or relative abundance in a plot along with its associated trait metrics. We will calculate each metric using all possible combinations of two traits up to all possible combinations of the maximum number of traits at each site. For example, at Sevilleta there are 10 different traits so there are 45 2-trait calculations, 120 3-trait calculations, 210 4-trait calculations, and so forth up to 10 9-trait calculations and 1 10-trait calculation. To measure

the effects of trait correlation on functional diversity, we will focus on metrics calculated with 4 traits only to standardize between sites. We will calculate the minimum, maximum, and mean correlation between the traits at each site.

## Statistical analyses

For each site separately, we will run mixed effects models to test the dependence of the three functional trait metrics on trait number and on trait correlation using the lme function from the nlme package in R [48]. To examine how trait number impacts the values of a given functional trait metric, we will run two models: Metric ~ trait number for 2–10 unique traits (the max number of traits at Cedar Creek and Sevilleta) and Metric ~ trait number for all traits possible to make sure our inferences are not impacted by excluding combinations of 11 and 12 traits at Konza. To examine how trait identity impacts the values of a given functional diversity metric, we will run 3 models for each site: Metric ~ min trait correlation, Metric ~ max trait correlation, Metric ~ mean trait correlation. We will explore which functional form of the predictor variables best fit the spread of the functional metric data by fitting linear, quadratic, cubic, and quartic fits. We will determine the models with the best fit using AIC values. We will account for repeated samples within plots by fitting plot as a random effect and an autoregressive correlation structure. We will account for multiple comparisons by adjusting our p-values using a Benjamini-Hochberg procedure [49].

## Timeline

All trait and community data have already been collected that will be used in this study, but none of the authors have analyzed any subset of the data in this way before.

We expect to finish cleaning data within 4 weeks of acceptance of the Registered Report Protocol. We will then complete the rest of the analyses and create figures over the following 6 weeks. We will finish writing the manuscript in another 6 weeks after data analysis is completed.

All code used for analyses will be uploaded to one of the author's OSF site before the second review stage.

## Supporting information

**S1 File.**
(DOCX)

## Author Contributions

**Conceptualization:** Timothy Ohlert, Kaitlin Kimmel, Meghan Avolio, Cynthia Chang, Benjamin Gerstner, Kimberly Komastu, Kenneth Whitney.

**Data curation:** Elisabeth Forrestel, Benjamin Gerstner, Sarah E. Hobbie, Peter Reich, Kenneth Whitney.

**Formal analysis:** Kaitlin Kimmel.

**Funding acquisition:** Elisabeth Forrestel, Sarah E. Hobbie, Peter Reich, Kenneth Whitney.

**Methodology:** Timothy Ohlert, Kaitlin Kimmel, Meghan Avolio, Cynthia Chang, Kimberly Komastu.

**Project administration:** Timothy Ohlert, Kaitlin Kimmel.

**Writing – original draft:** Timothy Ohlert, Kaitlin Kimmel.

**Writing – review & editing:** Meghan Avolio, Cynthia Chang, Elisabeth Forrestel, Benjamin Gerstner, Sarah E. Hobbie, Kimberly Komastu, Peter Reich, Kenneth Whitney.

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
