## [Decision Letter · Decision Letter 0]

15 Sep 2021

PONE-D-21-22835Exploring the impact of trait number and type on functional diversity metrics in real-world ecosystemsPLOS ONE

Dear Dr. Ohlert,

Thank you for submitting your manuscript to PLOS ONE. After careful consideration, we feel that it has merit but does not fully meet PLOS ONE’s publication criteria as it currently stands. Therefore, we invite you to submit a revised version of the manuscript that addresses the points raised during the review process.

Both reviewers agree that the idea described in the Registered Protocol to analyze methods using field data combining traits and scales of organization is interesting and novel and I also agree with this. The approach proposed is very interesting and very much needed yet both reviewers have raised some issues regarding the approach and methods planned to be used in the subsequent manuscript. Reviewers request a more thorough discussion on the limitations of the approach, which I agree with, e.g. comparing functional richness among studies with different sampling efforts and scale and their potential impacts on the outcome or the use of Gower distances that could be influenced by the inclusion of categorical traits or the use of the FD package (two-dimensionless) instead of alternative methods that allow to include multiple dimensions in the trait space, among other issues. There is also the potential within the manuscript to discuss why the authors think the FD package is the best methodology for the proposed study.

We look forward to receiving your revised manuscript.

Kind regards,

Iván Prieto Aguilar, Ph.D.

Academic Editor

PLOS ONE

 “KW - DEB-1257965 National Science Foundation Division of Environmental Biology https://www.nsf.gov/div/index.jsp?div=DBI

SH - DBI- 1725683, DEB-1753859, DEB- 1831944 National Science Foundation Division of Environmental Biology, National Science foundation Division of Biological Infrastructure https://www.nsf.gov/div/index.jsp?div=DBI

PR - DBI- 1725683, DEB-1753859 National Science Foundation Division of Environmental Biology, National Science foundation Division of Biological Infrastructure https://www.nsf.gov/div/index.jsp?div=DBI

EF - DEB-0841917 National Science Foundation Division of environmental biology https://www.nsf.gov/div/index.jsp?div=D”

Reviewers' comments:

Reviewer's Responses to Questions

**Comments to the Author**

1. Does the manuscript provide a valid rationale for the proposed study, with clearly identified and justified research questions?

Reviewer #1: Partly

Reviewer #2: Partly

2. Is the protocol technically sound and planned in a manner that will lead to a meaningful outcome and allow testing the stated hypotheses?

Reviewer #1: Partly

Reviewer #2: Partly

3. Is the methodology feasible and described in sufficient detail to allow the work to be replicable?

Reviewer #1: Yes

Reviewer #2: Yes

4. Have the authors described where all data underlying the findings will be made available when the study is complete?

Reviewer #1: Yes

Reviewer #2: Yes

5. Is the manuscript presented in an intelligible fashion and written in standard English?

Reviewer #1: Yes

Reviewer #2: Yes

6. Review Comments to the Author

You may also provide optional suggestions and comments to authors that they might find helpful in planning their study.

Reviewer #1: The idea to analyse the existing methods using real- world data on different combinations of traits and/or scales of organisation sound interesting and currently demanding (i.e. Mammola et al. 2021). The main concern about this article is that the approach seems too simplistic for me both from theoretical and methodological approaches.

From a methodological perspective, it is true that FD package is one of the most famous tools to calculate FD, but in my opinion, it is due to the simplicity of the tool which makes that many researchers routinely use this method, without many theorical and/or mathematical consideration. The improve of the theoretical development of FD metrics has been accompanied by a proliferation of methods. In this regard, new tools have been developed to improve FD package. The FD package has many constraints that the authors overlook in the article and, from my point of view, should be taken into account or at least discussed, especially for a methodological article. For example, as the authors explain, the number of dimensions depends on the number of selected traits, however, If I´m not wrong, the FD package only uses two dimensions to calculate FD metrics, in this regard new tools have been developed to make a selection of the dimensions based on their redundancy (see de Bello et al. 2016; Maire et al. 2015; Gutiérrez-Cánovas et al. 2020). In addition, functional richness often increases as new organisms are included in a group, if we compare Frich among studies, we are assuming the same sampling effort or the same sampling protocol, which I think is not the case in this article. Thus, to compare among studies a randomization could be desirable (Mammola et al. 2021). Other constraint is the use of Gower distances with the FD package, as de Bello et al. 2020 explains “the Gower distance can however produce a multi-trait dissimilarity with a disproportional contribution of certain traits, particularly categorical traits and bundle of correlated traits reflecting similar ecological functions. Hence categorical traits will contribute more to the multi-trait dissimilarity”. In de Bellos´s article is well explained the constraints of use the Gower distance with FD package. In fact, I don´t really understand why the authors use Gower and not Euclidian distances because If I´m right there is only one categorical trait (growth form in Konza), why don´t remove it?

From a theorical perspective, in the last five years a plethora of more sophisticated methods has been developed to represent the functional diversity, I don´t really know why to use FD package is the best option I really miss a strong dissertation about why the selected FD indices are the best choice. For instance, Frich are very sensitive to outliers, the space within extreme values of a convex hull is assumed to be homogeneous and also strongly depends of the number of species and traits, so why Frich should be a good index to compare Functional diversity among different studies or for meta-analyses? Why Functional divergence and not Rao or functional dispersion? In summary, the introduction needs more strong theoretical framework and less generalizations (such as Lines 59-60, 62-63, 74-75).

References

de Bello, F., Botta‐Dukát, Z., Lepš, J., & Fibich, P. (2021). Towards a more balanced combination of multiple traits when computing functional differences between species. Methods in Ecology and Evolution, 12(3), 443-448.

de Bello, F., Carmona, C. P., Lepš, J., Szava‐Kovats, R., & Pärtel, M. (2016). Functional diversity through the mean trait dissimilarity: resolving shortcomings with existing paradigms and algorithms. Oecologia, 180, 933–940. https://doi.org/10.1007/s00442‐016‐3546‐0

Gutiérrez-Cánovas, C., Sánchez-Fernández, D., González-Moreno, P., Mateos-Naranjo, E., Castro-Díez, P., & Vilà, M. (2020). Combined effects of land-use intensification and plant invasion on native communities. Oecologia, 192(3), 823-836.

Maire, E., Grenouillet, G., Brosse, S., & Villeger, S. (2015) How many dimensions are needed to accurately assess functional diversity? A pragmatic approach for assessing the quality of functional spaces. Global Ecology and Biogeography, 24, 728-740.

Mammola, S., Carmona, C. P., Guillerme, T., & Cardoso, P. (2020). Concepts and applications in functional diversity. Functional Ecology.

Reviewer #2: General Comments

This is a Registered Report Protocol, which is a new publication type for me. The proposed work rests heavily on simulation-like trait calculations in three different sites to test how functional diversity metrics respond to the number and correlation of traits used in different grassland ecosystems.

I have some fairly straightforward stats concerns listed in the methods comments below. Largely, I am concerned with treating the sites together in the statistical modeling, given different scales of vegetation monitoring; different surveyors estimating the ever subjective visual cover / abundance; and the different sets of traits. I think if each site was separated and analyzed only, more nuanced could be derived from each site result in how the trait selection and overall community structure may influence functional diversity metrics. The main question is about trends in diversity metrics based on trait number, and that could be answered while keeping separate models.

In addition, this is a simulation-based study, and I feel that it is well-positioned to explore a wider set of questions given the rich data behind it. In particular, Lines 186 – 187 highlight an important component that would be fairly straightforward to code and include, and would dig concretely into some of the uncertainty around functional metrics. Without it, the study feels a little limited.

Introduction

Lines 30-31: I think lots of attention has been given to assumptions in this space, which you highlight in the next few sentences. I suggest removing this sentence or toning down the language.

Methods

The scale of monitoring differs between data sets, with Konza at 10m2 and the other two at 1m2. Does this likely impact diversity estimates and outcomes of this study? It would certainly impact species richness estimates. How have similar studies dealt with the issue (perhaps species diversity studies have dealt with this explicitly somehow)? Because you model all of the sites together in a single model, I find this concerning. Konza is uniquely large, so including site as a fixed effect does not cover scale of monitoring.

Given that you are looking for trends in the metrics within a site, perhaps splitting them into separate models, or standardizing the response values, may deal with this potential variation. Modeling them separately would also allow you to explore the different effect size of the trait number predictor between sites, the variation at different trait number levels within sites, and the role that your different sets of traits may play in influencing the trends.

I also feel that it is hard to compare visual cover, as it is such a subjective metric and can vary so highly, especially between dry and mesic communities. The methods mention that in at least evenness, relative abundance is used. Can it also be used in the FDis calculations? If the models are split by site, this might not be an issue.

Lines 214 – 216: I’m not sure what “characterize trait type” means here.

Lines 226 – 228: Models, ideally, should be defined a priori or at least based on clear ecological reasoning. Is there a particular ecological reason that you chose to test linear, log, and quadratic fits?

Lines 228 – 229: I must have misread, but I thought each plot only had one value? (Lines 142; 153-155; 163).

7. PLOS authors have the option to publish the peer review history of their article (what does this mean?). If published, this will include your full peer review and any attached files.

Reviewer #1: No

Reviewer #2: No

---

## [Author Response · Author response to Decision Letter 0]

16 Nov 2021

We have provided a detailed response to the reviewer's comments as a separate file labeled Response to Reviewers as instructed.

---

## [Decision Letter · Decision Letter 1]

22 Feb 2022

PONE-D-21-22835R1Exploring the impact of trait number and type on functional diversity metrics in real-world ecosystemsPLOS ONE

Dear Dr. Ohlert,

Thank you for submitting your manuscript to PLOS ONE. After careful consideration, we feel that it has merit but does not fully meet PLOS ONE’s publication criteria as it currently stands. Therefore, we invite you to submit a revised version of the manuscript that addresses the points raised during the review process.

ACADEMIC EDITOR: The manuscritp was sent out for review to two reviewers that did not revise the previous version. Both reviewers again agree that the idea described in the Registered Protocol to analyze methods using field data combining traits and scales of organization is interesting and novel but one reviewer raised some issues regarding the cross site comparison and that many of the traits come from different treatments and different sets of traits are used in each site. The discussion on why using indexes in the FD package is now clearer and the authors propose to use alternative methods to Gower distances and including n-dimension hypervolumes (please check reviewer's 2 comments on this specific point). I have recommended minor revisions at this stage but please keep in mind when revising the manuscripts that reviewer's suggestions, specifically reviewer's 2 suggestions should be incorporated in full.==============================

We look forward to receiving your revised manuscript.

Kind regards,

Iván Prieto Aguilar, Ph.D.

Academic Editor

PLOS ONE

Journal Requirements:

Additional Editor Comments:

The manuscritp was sent out for review to two reviewers that did not revise the previous version. Both reviewers again agree that the idea described in the Registered Protocol to analyze methods using field data combining traits and scales of organization is interesting and novel but one reviewer raised some issues regarding the cross site comparison and that many of the traits come from different treatments and different sets of traits are used in each site. The discussion on why using indexes in the FD package is now clearer and the authors propose to use alternative methods to Gower distances and including n-dimension hypervolumes (please check reviewer's 2 comments on this specific point). I have recommended minor revisions at this stage but please keep in mind when revising the manuscripts that reviewer's suggestions, specifically reviewer's 2 suggestions should be incorporated in full.

Reviewers' comments:

Reviewer's Responses to Questions

**Comments to the Author**

1. Does the manuscript provide a valid rationale for the proposed study, with clearly identified and justified research questions?

Reviewer #3: Yes

Reviewer #4: Partly

2. Is the protocol technically sound and planned in a manner that will lead to a meaningful outcome and allow testing the stated hypotheses?

Reviewer #3: Yes

Reviewer #4: No

3. Is the methodology feasible and described in sufficient detail to allow the work to be replicable?

Reviewer #3: Yes

Reviewer #4: No

4. Have the authors described where all data underlying the findings will be made available when the study is complete?

Reviewer #3: Yes

Reviewer #4: Yes

5. Is the manuscript presented in an intelligible fashion and written in standard English?

Reviewer #3: Yes

Reviewer #4: Yes

6. Review Comments to the Author

You may also provide optional suggestions and comments to authors that they might find helpful in planning their study.

Reviewer #3: I did not review the original version of the proposal, but the current version seems fine. I have a few minor comments:

Introduction

L57-60 That is one argument (in addition to tractability) for using single trait indices, rather than multi trait indices. See Butterfield and Suding 2009 JEcol for one example.

L75-89 It seems like somewhere in the intro their ought to be a discussion of dimensionality reduction, which is standard practice in calculating many indices, e.g. in the FD library in R, and is noted in the Methods. Where does this fit in WRT the questions being addressed in this study?

Methods

L188 dC13?

L194 Reference? E.g. Siefert et al. 2015 EcoLetts or Jung et al. 2010 JEcol?

L208-209 extra ‘then’

Reviewer #4: Overall:

Right now I think you have some methodological hiccups that need to be ironed out, but those are all fixable and will become super obvious once you start coding. The writing needs to go into a bit more detail, particularly on the ecology of what the metrics mean and what they can be useful for. You also need to go into detail on what new insight this brings (why doing this on a non-simulated dataset is important). But I think the big problem with this is that your traits selection is confounded with site, limiting any cross-site comparisons. This might not be such a big deal if you had more sites, but with only 3 it seems like you won’t be able to say anything about the experimental vs natural systems. I worry that after all of your work all you’ll be left with is a statement saying that adding more traits doesn’t matter much if those traits are correlated. I do think the overall goals of the study are valuable, but I’m not convinced that this is the right match of question and dataset (at least as it stands).

Abstract

-Abstract could be improved (vague)

Intro:

-Some vagueness.

-Could bolster number of refs for key points (not needed of course)

-I think more emphasis could be placed on the importance of doing this work in real communities. I think this is a really cool selling point of this work and I think that the comparisons with some of the other work using e.g. simulations will make this a very useful and compelling study. As such, maybe devote a paragraph or so to this (perhaps between the current lines 74 and 75?)

-Measures of functional diversity: a bit more context linking the metrics you use with what they capture (in terms of ecology) could be useful. Perhaps a table?

Methods:

-3 sites (2 natural, 1 experimental)

-Glad to see that the trait data were recorded at each site

-Many of the traits come from different treatments than they are being used as proxies for. Definitely problematic.

-Different sets of traits at each site. Also problematic.

-Traits include individual-level (or organ level?), species-level, and population-level(?) but aggregated to species level.

- “Before analysis, we will drop traits with less 198 than 80% coverage of species by abundance in the community”: won’t you need to drop any trait with less than 100% coverage? Or else you can drop species without 100% trait coverage. These distance metrics require a complete set of traits.

-There are some issues and errors in the section on metrics (200-225).

--You give the abbreviated names (e.g. FDis) but not the full names (E.g. Functional Dispersion). You also don’t say what any of these metrics means from an ecological point of view.

--The five metrics don’t use PCoA (although you can certainly do them on a PCoA’ed data set). But you can also use them with z-scaled data or any other distances you’d like. I would think it would be preferable for this study to omit the PCoA, though. Or perhaps do it both ways.

--Your description of hypervolume calculations rests on one particular type of hypervolume. I agree that KDE is a good choice, but be careful not to equate hypervolumes with only one particular method. Also, it would be useful to explain why you choose KDE (probably one sentence), as well as why you prefer abundance weighted (presumably for consistency with other methods).

-Doesn’t PCoA require continuous values?

-Lines 226 - 239: In the previous paragraph you mention PCoA, but here you seem to be focusing on particular traits. Are you planning on applying a PCoA to each set of traits? Or am I missing something? Perhaps some quick clarification is needed.

Timeline:

-Might be a bit optimistic on data cleaning timeline (speaking from personal experience )

7. PLOS authors have the option to publish the peer review history of their article (what does this mean?). If published, this will include your full peer review and any attached files.

Reviewer #3: No

Reviewer #4: No

---

## [Author Response · Author response to Decision Letter 1]

22 Jul 2022

Dear PLOS ONE editor,

Re: Manuscript ID: PONE-D-21-22835R1

We thank you for your continued interest in our registered report and we are grateful for the opportunity to resubmit a revised manuscript. We thank you and the two additional reviewers for providing suggestions that will improve this manuscript. Both reviewers agreed that this study will be a valuable resource for ecological studies using trait-based functional diversity measures and that by increasing the scope of this study, we can make this contribution even more valuable.

In our revision, we addressed all comments raised by reviewers. In particular, both reviewers asked for greater clarity regarding the metrics being used including the dimensionality to be used when calculating these metrics. We have added considerable text explaining the metrics and parameters with which they will be calculated. In addition, we have added a table that organizes the metrics, abbreviations, ecological meaning, and their use in the literature. We believe this table adds necessary background and improves the logic for readers.

We also responded to the other comments from the reviewers in the text that follows.

We hope that this revised manuscript answers your concerns, and we are grateful for the helpful feedback. We think that the revised manuscript is a great improvement.

Thank you in advance for considering our resubmission.

Sincerely,

Timothy Ohlert and Kaitlin Kimmel

Reviewer #3: I did not review the original version of the proposal, but the current version seems fine. I have a few minor comments:

Introduction

L57-60 That is one argument (in addition to tractability) for using single trait indices, rather than multi trait indices. See Butterfield and Suding 2009 JEcol for one example.

We agree with the importance of single trait indices and have added text in the introduction to reflect this (lines 69-72). 

L75-89 It seems like somewhere in the intro their ought to be a discussion of dimensionality reduction, which is standard practice in calculating many indices, e.g. in the FD library in R, and is noted in the Methods. Where does this fit in WRT the questions being addressed in this study?

We agree that dimensionality is an important factor to consider when calculating FD metrics - and one that is often not discussed in methods sections when the FD package is deployed. We have now added text about this in lines 109-220: “Further, FD metric calculation in high dimensional space can require dimensionality reduction – another decision that can impact the value of the metric calculated”

In our study, we are calculating metrics using two to twelve traits. Similar to Legras et al 2020, we are going to hold the number of dimensions equal to 2 for our analyses. This will only impact the calculation of functional richness (FRich) as the other metrics do not require dimensionality reduction (note: FRich does not either if all traits are continuous, but we have several categorical traits in our dataset). We will then conduct a sensitivity analysis to determine if holding the number of dimensions equal to 3, 4, and the maximum (dimensions = number of traits when using all continuous traits or dimensions = number of traits -1 when including categorical traits) produce similar results. We now explicitly mention this in lines 280-288

Methods

L188 dC13? 

Yes. We have made that correction.

L194 Reference? E.g. Siefert et al. 2015 EcoLetts or Jung et al. 2010 JEcol?

We have added these references along with Bolnick et al. 2011 (Trends in ecology & evolution) and Westerband et al. 2021 (Annals of botany) on intraspecific variation. 

L208-209 extra ‘then’

We have made that correction

Reviewer #4: Overall:

Right now I think you have some methodological hiccups that need to be ironed out, but those are all fixable and will become super obvious once you start coding. The writing needs to go into a bit more detail, particularly on the ecology of what the metrics mean and what they can be useful for. You also need to go into detail on what new insight this brings (why doing this on a non-simulated dataset is important). But I think the big problem with this is that your traits selection is confounded with site, limiting any cross-site comparisons. This might not be such a big deal if you had more sites, but with only 3 it seems like you won’t be able to say anything about the experimental vs natural systems. I worry that after all of your work all you’ll be left with is a statement saying that adding more traits doesn’t matter much if those traits are correlated. I do think the overall goals of the study are valuable, but I’m not convinced that this is the right match of question and dataset (at least as it stands).

In a previous version, another reviewer correctly pointed out that statistical tests involving just these three sites would be inappropriate for reasons that likely concern reviewer #4. Our goal is not to attempt a thorough metanalysis, since site-specific data for this number of traits in similar ecosystems has yet to be compiled on a global scale. Instead, we are taking a smaller step by applying analysis of FD metrics to real ecosystems and trait data, and adding confidence to our conclusions with the addition of experiments with high-quality data.

Abstract

-Abstract could be improved (vague)

We have updated the abstract to be more specific on the goals of our studies, where it fits within the existing literature, and our methods. We, of course, will update the abstract once we have results for the next phase of the Registered Report. 

Intro:

-Some vagueness.

We have added more detail specifically around ecological theory and the relevance of analyzing real community and trait data (lines 84-95 and throughout introduction).

-Could bolster number of refs for key points (not needed of course)

We have increased the number of references overall, including for some of our main points for which we also added additional text.

-I think more emphasis could be placed on the importance of doing this work in real communities. I think this is a really cool selling point of this work and I think that the comparisons with some of the other work using e.g. simulations will make this a very useful and compelling study. As such, maybe devote a paragraph or so to this (perhaps between the current lines 74 and 75?)

We agree that the focus on real trait data from real communities is an important aspect of this study. We have added text in the introduction to highlight this (lines 84-95). 

“Calculating functional diversity measures in natural communities poses additional challenges both ecological and practical. Real plant communities are non-random assemblages of species which are influenced by competitive interactions, coexistence, mutualisms, niche partitioning, and environmental filtering among many other processes of community assembly [20,21,22,23,24,25]. Functional diversity metrics are likely to exhibit patterns due to ecologically meaningful correlation of traits in real communities, in particular, among suites of traits typically used in community ecology such as the leaf economic spectrum and root economic spectrum [26,27]. Moreover, real data collection introduces constraints on trait data, such as realistic numbers of traits collected given limited resources and missing trait data, particularly for rare species. Functional diversity metrics, therefore, are most often calculated with fewer traits and fewer species than those in studies based on simulated communities.”

-Measures of functional diversity: a bit more context linking the metrics you use with what they capture (in terms of ecology) could be useful. Perhaps a table?

We have taken this recommendation and added a table including information about the list of functional diversity indices.

Methods:

-3 sites (2 natural, 1 experimental)

-Glad to see that the trait data were recorded at each site

-Many of the traits come from different treatments than they are being used as proxies for. Definitely problematic. 

The traits from Sevilleta are collected under ambient conditions (see lines 2020-2022) and the communities are also in ambient conditions (lines 190-192). The traits from Konza come from a watershed that is burned annually, but we do not think that the potential difference in these traits will impact the relationship between number or correlation of traits and metric magnitude. The traits from Cedar Creek come from monocultures that are subject to the same CO2 and nitrogen treatments as the communities being analyzed.

-Different sets of traits at each site. Also problematic.

The reviewer is correct that it would be inappropriate to include all sites in a single model given differences in traits, however, each site will be handled independently as three separate analyses. The objective of this study is not to directly compare the magnitude of FD metrics between sites but rather, to look for patterns in the direction of the slope as the number of traits included in the metric increases or as traits become more correlated. Thus, our analyses are agnostic of which traits are used to calculate the FD metrics other than their correlation with each other. The purpose of using three sites is to see if there are similar patterns within sites rather than drawing conclusions from just one site. 

-Traits include individual-level (or organ level?), species-level, and population-level(?) but aggregated to species level.

We do include traits collected at these different levels - this is typically of analyses in the trait literature (Frenette-Dussault et al. 2012 Jecol, Biswas & Mallik 2010, Kimmel et al. 2019) and is even preferable in order to encompass the dimensionality of plant form and function (Laughlin 2013). We have added text to clarify this under the Trait Data section (lines 210-211).

- “Before analysis, we will drop traits with less 198 than 80% coverage of species by abundance in the community”: won’t you need to drop any trait with less than 100% coverage? Or else you can drop species without 100% trait coverage. These distance metrics require a complete set of traits.

It seems our wording of this section has caused confusion. We are dropping species that do not have 100% trait coverage, but we still want to make sure that what coverage we do have is at least 80% of the total species abundance. Thus, we may have a reduced community, but still have 100% coverage by species. We don’t have a priori knowledge of what percentage of species will be dropped from the analysis because we have not looked at the data yet. We will report this number in the final draft after analysis is done. The reviewer is correct that 100% trait coverage is important for calculating these metrics and, therefore, it is common in such studies to focus on common species with full trait coverage and exclude rare species which often have incomplete trait coverage. Though this approach de-emphasizes the contribution of rare species to functional diversity, most of the metrics we use are abundance weighted anyway which also de-emphasizes rare species.

We will be characterizing the dominant species in the community - so we may be misrepresenting the community where rare species play an important part in ecosystem function (e.g. Dee et al 2019 TREE). However, when ecologists are performing trait analyses on data by combining the TRY dataset with their local communities, there may not be full trait coverage. We are grappling with this disparity and an unfortunate reality of trait analyses by removing traits that do not characterize 80% or more of the community. 

-There are some issues and errors in the section on metrics (200-225).

--You give the abbreviated names (e.g. FDis) but not the full names (E.g. Functional Dispersion). You also don’t say what any of these metrics means from an ecological point of view.

We now put the abbreviations in table 1 (new), in the introduction paragraph, and then provide the full names the first time we mention them in the methods. In addition, table 1 briefly explains the ecological relevance of each metric and provides examples of usage in the literature. 

--The five metrics don’t use PCoA (although you can certainly do them on a PCoA’ed data set). But you can also use them with z-scaled data or any other distances you’d like. I would think it would be preferable for this study to omit the PCoA, though. Or perhaps do it both ways.

While FRich, FDis, FDiv, FEve, and Rao’s Q do NOT require dimensionality reduction via PCoA, but the FD package does reduce the dimensionality via PCoA. Except for FRich, we can set the number of dimensions equal to the number of traits to preserve the trait axes when using this function. We understand that this is another layer of complexity we are adding into our study, but it is an important aspect of conducting trait analyses that we do not wish to gloss over. We explain this further below. 

--Your description of hypervolume calculations rests on one particular type of hypervolume. I agree that KDE is a good choice, but be careful not to equate hypervolumes with only one particular method. Also, it would be useful to explain why you choose KDE (probably one sentence), as well as why you prefer abundance weighted (presumably for consistency with other methods).

We have clarified which KDE metrics we are testing (see table 100, improved the nomenclature (line 240), and added details on how these metrics are calculated (lines 278-280). Additionally, we have added text explaining why KDE hypervolumes, along with metrics from the FD package, are the best choice (lines 264-270).

“All KDE-based functional diversity metrics will be calculated using the hypervolume and bat packages in R [34,48]. KDE richness is the total volume of the n-dimensional hypervolume created from unweighted trait values present in the community. KDE evenness is the overlap between the abundance-weighted n-dimensional hypervolume and a similar hypervolume in which all traits and abundances are distributed evenly. KDE dispersion is the average distance between random points within the n-dimensional hypervolume and the hypervolume centroid.“

-Doesn’t PCoA require continuous values?

The PCoA is done on the species-species distance matrix, and the categorical variables are taken into consideration in the creation of the distance matrix which is done using Gower dissimilarity. Thus, the PCoA is done on the continuous distance values rather than on the raw traits. We add this detail in line 282-290. 

-Lines 226 - 239: In the previous paragraph you mention PCoA, but here you seem to be focusing on particular traits. Are you planning on applying a PCoA to each set of traits? Or am I missing something? Perhaps some quick clarification is needed.

We will conduct a PCoA on each set of traits as if we were characterizing our community on those traits, so on each set of traits. In our study, we are calculating metrics using two to twelve traits. See lines 282-290.

“The categorical variables are taken into consideration in the creation of the distance matrix which is done using Gower dissimilarity. Thus, the PCoA is done on the continuous distance values rather than on the raw traits. Similar to Legras et al. [19], we are going to hold the number of dimensions equal to 2 for only our calculation of functional richness (FRich) as the other metrics do not require dimensionality reduction (note: FRich does not either if all traits are continuous, but we have several categorical trats in our dataset). We will then conduct a sensitivity analysis to determine if holding the number of dimensions equal to 3, 4, and the maximum (dimensions = number of traits when using all continuous traits or dimensions = number of traits-1 when including categorical traits) produce similar results.”

Timeline:

-Might be a bit optimistic on data cleaning timeline (speaking from personal experience)

We have created a new timeline given that both lead authors have taken new jobs and will need more time to complete analyses.

---

## [Editor Report · Decision Letter 2]

27 Jul 2022

Exploring the impact of trait number and type on functional diversity metrics in real-world ecosystems

PONE-D-21-22835R2

Dear Dr. Ohlert,

We’re pleased to inform you that your manuscript has been judged scientifically suitable for publication and will be formally accepted for publication once it meets all outstanding technical requirements.

Kind regards,

Iván Prieto Aguilar, Ph.D.

Academic Editor

PLOS ONE

Additional Editor Comments (optional):

I would like to remark that the authors have done a great job incorporating reviewer's comments and adjusting timelines for having the full manuscript ready. The trait data collection is impresive and, althgouh comparing sites will be a challenge, the incorporation of new sites in the future will probably open doors to this comparison.
---

## [Editor Report · Acceptance letter]

2 Aug 2022

PONE-D-21-22835R2 

Exploring the impact of trait number and type on functional diversity metrics in real-world ecosystems 

Dear Dr. Ohlert:

I'm pleased to inform you that your manuscript has been deemed suitable for publication in PLOS ONE. Congratulations! Your manuscript is now with our production department. 

Kind regards, 

on behalf of

Dr. Iván Prieto Aguilar 

Academic Editor

PLOS ONE